# Efficacy of Heterologous Prime-Boost Vaccination with H3N2 Influenza Viruses in Pre-Immune Individuals: Studies in the Pig Model

**DOI:** 10.3390/v12090968

**Published:** 2020-09-01

**Authors:** Sharon Chepkwony, Anna Parys, Elien Vandoorn, Koen Chiers, Kristien Van Reeth

**Affiliations:** 1Laboratory of Virology, Faculty of Veterinary Medicine, Department of Virology, Parasitology and Immunology, Ghent University, 9820 Merelbeke, Belgium; Sharon.Chepkwony@UGent.be (S.C.); Anna.Parys@UGent.be (A.P.); Elien.Vandoorn@UGent.be (E.V.); 2Laboratory of Veterinary Pathology, Faculty of Veterinary Medicine, Ghent University, 9820 Merelbeke, Belgium; Koen.Chiers@UGent.be

**Keywords:** influenza, swine, humans, pre-existing immunity, vaccination, heterologous prime-boost, H3N2, antibody cross-reactivity

## Abstract

In a previous study in influenza-naïve pigs, heterologous prime-boost vaccination with monovalent, adjuvanted whole inactivated vaccines (WIV) based on the European swine influenza A virus (SwIAV) strain, A/swine/Gent/172/2008 (G08), followed by the US SwIAV strain, A/swine/Pennsylvania/A01076777/2010 (PA10), was shown to induce broadly cross-reactive hemagglutination inhibition (HI) antibodies against 12 out of 15 antigenically distinct H3N2 influenza strains. Here, we used the pig model to examine the efficacy of that particular heterologous prime-boost vaccination regimen, in individuals with pre-existing infection-immunity. Pigs were first inoculated intranasally with the human H3N2 strain, A/Nanchang/933/1995. Seven weeks later, they were vaccinated intramuscularly with G08 followed by PA10 or vice versa. We examined serum antibody responses against the hemagglutinin and neuraminidase, and antibody-secreting cell (ASC) responses in peripheral blood, draining lymph nodes, and nasal mucosa (NMC), in ELISPOT assays. Vaccination induced up to 10-fold higher HI antibody titers than in naïve pigs, with broader cross-reactivity, and protection against challenge with an antigenically distant H3N2 strain. It also boosted ASC responses in lymph nodes and NMC. Our results show that intramuscular administration of WIV can lead to enhanced antibody responses and cross-reactivity in pre-immune subjects, and recall of ASC responses in lymph nodes and NMC.

## 1. Introduction

All swine influenza A viruses (SwIAV) of the H3N2 subtype have their hemagglutinin (HA) and neuraminidase (NA) proteins derived from H3N2 viruses that once circulated in humans [1,2]. Repeated transmissions of distinct human-origin viruses to swine have led to an increase in the genetic diversity of SwIAV globally, with strains being confined to their geographical regions of introduction [3]. For instance, H3N2 SwIAV that circulate in European pigs were derived from a human virus that was circulating in the mid-1970s [4]. On the other hand, H3N2 SwIAV that circulate in North American pigs were derived from antigenically distinct human viruses that circulated in the 1990s and in the recent past, in 2011 [1]. This has resulted in the existence of distinct lineages and clusters of H3N2 SwIAV within the same region. These SwIAV have continued to evolve independently from their human counterparts, through either antigenic drift (mutations occurring in genes) or genetic reassortment (mixing of genes in case of simultaneous infections with different strains) with established strains. The rate of evolution of the European SwIAV is much slower as compared with that of the North American SwIAV as well as the human influenza viruses [5,6]. Because variants of human H3N2 influenza viruses keep replacing each other over time, contemporary human H3N2 viruses are genetically and antigenically distinct from those that circulate in swine. Thus, swine serve as reservoirs for the past human viruses, which occasionally jump back to humans [7,8,9,10]. The existence of multiple drift variants of SwIAV poses a great challenge to the effectiveness of vaccines for swine, just as continuous antigenic drift of human influenza viruses does for the human vaccines. In addition, there are no vaccines that can protect humans against zoonotic infections with SwIAV and swine against human influenza viruses. There is, therefore, a strong need for vaccines that can elicit potent and broadly protective immunity against drift variants of influenza viruses of both swine and humans.

Heterologous prime-boost vaccination is one strategy that can broaden immune responses to influenza viruses. In studies in humans, poultry, mice, and ferrets, using antigenically distinct strains for primary and booster vaccinations was shown to elicit broad antibody responses against drifted influenza strains within the same HA subtype [11,12,13,14,15,16]. In a previous study, we injected pigs with adjuvanted whole inactivated vaccines (WIV) based on a European H3N2 SwIAV (A/swine/Gent/172/2008, G08) followed 4 weeks later by a North American SwIAV (A/swine/Pennsylvania/A01076777/2010, PA10). This approach stimulated hemagglutination inhibition (HI) antibodies that reacted with 80% of a panel of 15 antigenically distinct H3N2 influenza viruses of both swine and humans. The heterologous prime-boost vaccination strategy was more potent than two administrations of matched bivalent vaccine or homologous monovalent vaccine. The latter strategy induced antibodies against less than 40% of the tested viruses [17].

While most studies are carried out in influenza-naïve animals, humans encounter numerous influenza strains by either infection or vaccination during their lifetime [18,19]. Therefore, most humans have pre-existing immunity to one or several influenza viruses. The effects of pre-existing immunity on responses against vaccination or subsequent infections can either be negative or positive. These effects have been described using the concepts of original antigenic sin (OAS) and immune imprinting or antigenic seniority [20,21,22,23].

Influenza infections in pigs and humans are not only caused by the same H1N1 and H3N2 subtypes of influenza viruses, the resulting clinical disease and pathogenesis are also very similar in both species [24,25,26]. We are using the pig as a model to examine vaccination strategies that can offer broader protection in both pigs and humans. The results of our previous heterologous prime-boost vaccination study [17] aroused our interest in examining the impact of pre-existing immunity on that particular vaccination strategy, given that the results were promising. To this purpose we first infected pigs with a human H3N2 virus that circulated more than 25 years ago, A/Nanchang/933/1995 (NC95). Because most humans are exposed to influenza viruses by the age of 6 [27,28], it is likely that people over the age of 30 have been exposed to the prototype H3N2 influenza virus strain that circulated in 1995. Subsequently, we injected the pigs with adjuvanted WIV based on two antigenically distinct strains: G08 followed by PA10 and vice versa (heterologous prime-boost vaccination). Homologous prime-boost vaccination groups were included for comparison purposes. We observed that pigs with pre-existing immunity to NC95 had enhanced antibody responses after vaccination and complete or near complete protection after challenge with an antigenically distinct H3N2 strain. These results suggest a beneficial effect of pre-existing infection-immunity on subsequent exposures to WIV.

## 2. Materials and Methods

### 2.1. Viruses and Vaccines

Four different influenza viruses were used for immunization and/or challenge of pigs. The human virus, NC95, was used for infection. For vaccination, WIV vaccines were prepared from two H3N2 SwIAV: a European SwIAV, G08, and a North American cluster IV SwIAV, PA10. These two viruses were used in our previous heterologous prime-boost vaccination study in influenza-naïve pigs [17]. The challenge virus was A/swine/Missouri/A01840724/2015 (MO15), which belongs to the most recent antigenic cluster of H3N2 SwIAV in North America, the so-called “novel reassortant” H3N2 viruses. The HA of this virus is derived from a recent human virus that circulated between 2010 and 2011, while its NA is derived from the cluster IV North American H3N2 SwIAV. It is genetically and antigenically very distinct from the three viruses used for immunization (Table 1). The propagation of viruses and preparation of vaccines were performed as previously described [17]. Each vaccine dose contained 256 hemagglutinating units (HAU) in combination with 20% Emulsigen^®^, the commercial oil-in-water adjuvant (MVP Laboratories, NE, USA).

HA sequences of these viruses were downloaded from GenBank. HA1 amino acid sequences were aligned, and % amino acid (aa) identity as well as the number of different aa acids at putative antigenic sites, as described elsewhere [29], were determined in MEGA 7.0.26 software [30]

The viruses used in serological studies included swine and human influenza viruses with H3 or N2 that were antigenically distinct from the viruses used for inoculation (These viruses are shown in Tables 3 and 4 in the results section). North American SwIAV were obtained from the United States Department of Agriculture (USDA) swine influenza repository and from Dr. Christopher Olsen, Department of Pathobiological sciences, School of Veterinary Medicine, University of Wisconsin-Madison, USA. Human seasonal H3N2 and H3N2v viruses were obtained from Dr. John McCauley, Francis Crick World Influenza Centre, London, UK.

### 2.2. Experimental Design

Fifty-eight five-week-old conventional influenza-naïve pigs were used in two separate experiments. In the first and primary experiment that aimed to examine the impact of pre-existing immunity on serum antibody responses following vaccination, 30 pigs were inoculated as shown in Table 2. Five groups of five pigs were inoculated intranasally with 10^4.9^ TCID_50_ of the human NC95 strain, to mimic pre-existing immunity in the human population. Seven weeks after infection, four of the infected groups were vaccinated twice at a 4 week interval with different vaccine strains, G08 followed by PA10 or vice versa (heterologous prime-boost vaccination) or with the same vaccine strains, G08 or PA10 (homologous prime-boost vaccination). The fifth group was mock-vaccinated and served as the infection-immune challenge control group. A sixth group of pigs was mock-infected/vaccinated and served as the naïve challenge control group. Six weeks after the last vaccination, all the 30 pigs were challenged intranasally with 10^7^ TCID_50_ of MO15 in a volume of 5 mL phosphate-buffered saline (PBS). The pigs were euthanized 3 days post challenge, and nasal mucosa, trachea, and lung samples were collected for evaluation of virus titers. Trachea and lung samples were also used for histopathological examinations. Blood for sera was collected at the time of infection (week 0), 4 weeks later (week 4), at the time of each vaccination (weeks 7 and 11), 2 weeks after each vaccination (weeks 9 and 13), 4 weeks after the last vaccination (week 15), and at the time of challenge. Sera collected at all these time points, except at the time of challenge, were examined in HI assay for antibodies against the three strains used for inoculation (NC95, G08, PA10). Sera collected at weeks 9 and 13 were examined further in HI and enzyme-linked lectin assay (ELLA) for HI and neuraminidase inhibition (NI) antibodies, respectively, against broader panels of antigenically distinct H3 or N2 virus strains. Sera collected at the time of challenge were used to examine the correlation between HI antibody titers against MO15 and post-challenge virus titers in the lungs.

In the second experiment, which aimed to examine the impact of pre-existing immunity on antibody-secreting cell (ASC) and interferon-secreting cell (ISC) responses following vaccination, another group of 28 influenza-naïve pigs was used. Twenty-four pigs were inoculated as the group NC95-PA10-G08 in Table 2 above. This is because this vaccination regimen resulted in the best serological and challenge results overall in the first experiment of this study. Four non-inoculated pigs were included as controls. Pigs were sampled at weeks 1, 1.5 (day 11), 2, 7, 8, 11, and 12 after infection. Weeks 7 and 11 coincided with the time of the first and second vaccinations, respectively. At each of these time points except at day 11, four infection-immune vaccinated pigs were euthanized for collection of nasal mucosa (NMC) and lymph nodes, including tracheobronchial lymph nodes (TBLN) and retropharyngeal, parotid, and superficial cervical lymph nodes, which in this study we will call lymph nodes of the head (LNH). One control pig was euthanized at weeks 7, 8, 11, and 12 for the same purpose. Blood for sera and isolation of peripheral blood mononuclear cells (PBMC) was also collected at each of the above time points. Mononuclear cells from all the tissues were examined for IgG and IgA ASC responses against the viruses used for immunization and two antigenically distinct human H3N2 influenza viruses, A/Victoria/3/1975 (VIC75) and A/Hong Kong/4801/2014 (HK14), in enzyme-linked immunospot (ELISPOT) assay. Due to insufficient cell numbers, we were unable to examine IgG and IgA ASC responses in LNH at week 1, and IgG ASC responses in NMC at week 2. Also, due to insufficient number of cells in all the other tissues, ISC responses were only examined in PBMC, in ELISPOT assays against the same viruses mentioned above. Sera collected at each of the above time points, including an additional time point, week 9 (14 days post vaccination 2), were examined in HI assay for antibodies against the same viruses. The use of experimental animals was approved by the Ethical and Animal Welfare Committee of the Faculty of Veterinary Medicine of Ghent University (project identification code: 2018-87, approval date: 21st January 2019).

### 2.3. Serological Assays

Hemagglutination inhibition assays were performed according to standard procedures using turkey erythrocytes (WHO, 2011). The enzyme-linked lectin assay was also performed as previously described [31]. Briefly, heat-inactivated serum samples were pretreated with receptor-destroying enzyme (RDE, Sigma-Aldrich, St. Louis, MO, USA) at 37 °C for 18 h. RDE was then inactivated by incubating the samples at 56 °C for 8 h. Before use, the samples were clarified by centrifugation at 20,000× *g* for 30 min. Serially diluted sera were added to 96-well plates precoated with fetuin substrate. Virus of predetermined concentration was then added and incubated at 37 °C for 18 h. Neuraminidase-induced removal of sialic acid from the substrate was measured by adding a peroxidase-conjugated lectin peanut agglutinin (PNA, Sigma-Aldrich, St. Louis, MO, USA) that specifically recognizes unmasked saccharides. The amount of bound PNA which correlates directly with the amount of sialic acid removed from the substrate as well as the neuraminidase activity was determined by analyzing sample absorbance at 490 nm.

### 2.4. Virus Titration and Lesion Scores

We evaluated virus titers in nasal swabs, and in 20% tissue homogenates of the respiratory part of the nasal mucosa, the trachea, and a pooled sample of the apical, cardiac, and diaphragmatic lobes of the right lung. Virus titration was performed in MDCK cells as previously described [32]. Virus titers were calculated using the Reed–Muench method [33]. Macroscopic and microscopic lesions of the lung and trachea were analyzed and scored as previously described [34].

### 2.5. Tissue Collection and Processing for ELISPOT Assays

Tissue collection and processing was conducted as previously described [35]. Briefly, PBMC were isolated from heparinized whole blood by centrifuging at 2100 rpm over Ficoll-Paque (GE healthcare Life, Uppsala, Sweden) for 45 min. The cells were then resuspended in RPMI 1640-GlutaMAX (Life Technologies, Paisley, UK) medium with 1% sodium pyruvate, 1% nonessential amino acids, 10% fetal calf serum, 1% penicillin–streptomycin, and 0.5% gentamicin (Complete RPMI medium). Lymph nodes of the head and TBLN were processed separately through a 60-mesh tissue screen (Sigma-Aldrich, St. Louis, MO, USA), filtered through a 100 μm strainer (VWR International, LLC, Radnor, PA, USA), and centrifuged over Ficoll-Paque, and cells were resuspended in complete RPMI medium. The NMC tissues, which included the mucosal lining and nasal associated lymphoid tissue, were digested with 100 mg of collagenase type IV (Life Technologies, Grand Island, NY, USA) and 1 mg of DNase (Roche Diagnostics GmbH, Mannheim, Germany) in 100 mL of RPMI 1640-GlutaMAX at 37 °C for 3 h. The resulting supernatant was filtered through a 70 μm strainer (VWR International, LLC, Radnor, PA, USA) and centrifuged over a Percoll (GE healthcare Life, Uppsala, Sweden) gradient of two specific gravities, 1.080 and 1.055. The isolated mononuclear cells were resuspended in complete RPMI medium.

### 2.6. ASC and ISC ELISPOT

IgG and IgA ASC responses in PBMC were determined using ELISPOT assay as previously described [17,36] except that for IgG ASC responses, monoclonal mouse anti-porcine IgG (MT421, MABTECH Inc., Cincinnati, OH, USA) was used as the capture antibody and biotinylated mouse anti-porcine IgG (MT424, MABTECH Inc., Cincinnati, OH, USA) as the detection antibody. IgG and IgA ASC responses in NMC, LNH, and TBLN were also examined as described above. ISC responses were only determined in PBMC, as previously described [37,38,39]. Briefly, 96-well ELISPOT filter plates (Merck Millipore, Molsheim, France. Cat. No. MAIPS4510) were coated overnight at 4 °C with monoclonal antibody to porcine IFN-γ (BD Pharmingen, San Jose, CA, USA, Cat. No. 559961) in PBS. PBMC suspended in RPMI 1640-GlutaMAX were added to the plates at a concentration of 5 × 10^5^ cells per well in duplicates. Thereafter, plain medium or media containing one of five viruses (G08, PA10, NC95, HK14, and VIC75) diluted to one multiplicity of infection (MOI) or concanavalin A (ConA, Merck KGaA, Darmstadt, Germany) were added. ConA was used as a positive control while plain medium was used as a negative control for ISC responses. A lower concentration of cells (5 × 10^4^ cells per well) was added to wells with ConA. The plates were incubated for 18 h at 37 °C in 5% CO_2_. They were then washed with PBS and blocked with complete RPMI medium for 2 h at 37 °C. Thereafter, they were washed with PBS containing Tween 20, incubated with biotinylated monoclonal antibodies to porcine IFN-γ (BD Pharmingen, San Jose, CA, USA Cat. No. 559958) for 2 h at room temperature, washed again, and incubated with HRP-conjugated streptavidin (Invitrogen, Carlsbad, CA, USA) for 1 h at room temperature. Following a final wash, the plates were incubated with TMB membrane substrate (Sigma-Aldrich, St. Louis, MO, USA) for 10 min at room temperature, and rinsed with distilled water. The average numbers of spots in wells without virus were subtracted from the average number in the virus-specific wells. The numbers of ISCs cells were reported as the number of spots per million PBMC.

### 2.7. Statistics

Nonparametric Mann–Whitney U test was used to compare antibody titers between groups, while analysis of variance (ANOVA) was used to compare virus titers. Nonparametric Spearman correlation coefficients were determined in GraphPad Prism 6 (Graphpad Software Inc., San Diego, CA, USA). Antibody titers below the detection limit (<10) were assigned a value of 5, while virus titers below the detection limit (1.7 log_10_ TCID_50_) were assigned a value of 0.85 log_10_ TCID_50_.

## 3. Results

### 3.1. Serum Antibody Responses and Protection against Challenge after Vaccination of Infection-Immune Pigs

We examined the breadth of serum HI and NI antibody responses and protection against challenge after heterologous prime-boost vaccination of infection-immune pigs.

#### 3.1.1. Serum Antibody Responses against the Strains Used for Infection and Vaccination

To examine the evolution of antibodies against the viruses used for infection and vaccination (NC95, G08, and PA10), sera collected at weeks 0, 4, 7, 11, and 15 were tested for antibody responses in HI assays. Figure 1 shows the geometric mean HI antibody titers (GMT) against each of the three virus strains in each of the five infection-immune groups. Pigs in all groups tested negative for antibodies against the three viruses at week 0. The naïve challenge control group tested negative at all time points. Antibody responses against NC95 (Figure 1A) but not against G08 (Figure 1B) or PA10 (Figure 1C) were observed at week 4 post infection in all the five infection-immune groups. By week 7 (the time of first vaccination), the titers against NC95 had become 4- to 8-fold lower. At week 11, the effects of the first vaccination were evident. All the groups, except the infection-immune challenge control group (NC95-PBS-PBS), showed significant increase in anti-NC95 antibody titers. At this time point, responses against G08 and PA10 were also observed. Anti-G08 antibody titers were significantly lower than anti-NC95 antibody titers in all groups except in the NC95-G08-G08. Likewise, anti-PA10 antibody titers were significantly lower than anti-NC95 titers in all groups except in the NC95-PA10-PA10 group. The results at week 15 show that responses against the three viruses were boosted following the second vaccination in all the four infection-immune vaccinated groups except for anti-PA10 antibody responses in the NC95-G08-G08 group. The increase in antibody titers against the three viruses in all the four groups, however, were not statistically significant except for anti-G08 titers in the NC95-PA10-G08 group. There were no significant differences in the responses against the three strains between groups that received homologous and heterologous vaccines (*p* > 0.05) except for lower anti-G08 titers (Week 11) and higher anti-PA10 responses (Weeks 11 and 15) in the NC95-PA10-G08 group than in the NC95-G08-G08 group (*p* < 0.05).

#### 3.1.2. Serum Antibody Responses against Antigenically Distinct Influenza Strains

To examine the breadth of antibody responses, sera collected 2 weeks after each vaccination (Weeks 9 and 13) were examined for HI and NI antibodies against panels of 18 H3N2 and 10 H3N2 or H1N2 influenza virus strains, respectively. These test viruses were selected based on their antigenic diversity with the vaccine strains as described in our previous study [17] and also included the strains used for immunization. In the current study, we included three additional H3N2 strains that were not examined in our previous study in HI assays. Two of these strains belong to the novel cluster of SwIAV viruses (A/swine/Missouri/A01476459/2012 and A/swine/Missouri/A01840724/2015) that currently circulate in North America and one is a more recent human virus, A/Hong Kong/4801/2014. Of these, only A/swine/Missouri/A01840724/2015 was included in the ELLA assays.

Geometric mean HI antibody titers against the tested viruses are shown in Table 3. The naïve challenge control group was negative for HI antibodies at both time points. In humans, an HI antibody titer ≥40 against a particular influenza virus is generally considered sero-protective against that virus [40,41]. In the infection-immune challenge control group, a mean antibody titer ≥40 was observed only against NC95 at both time points. After the first vaccination (Table 3, Week 9), all four infection-immune vaccinated groups developed titers ≥40 against 12 out of the 18 viruses. Antibody titers were lowest against five (VIC75, WI05, PER09, VIC11, HK14) out of seven human viruses and one virus of the European swine lineage, ENG87. The second vaccination resulted in further broadening of antibody responses (Table 3, Week 13). The number of viruses against which a titer ≥40 was achieved increased from 12 to 13 or 14 in three groups (last row of the table). Antibody titers against human H3N2 viruses isolated between 2005 and 2014 (WI05 to HK14) remained <40. In the NC95-PA10-PA10 group, the number of viruses against which a titer of 40 was achieved remained the same, but antibody titers increased as compared with those after the first vaccination. Heightened HI antibody titers were also observed in the group NC95-PA10-G08. HI antibody responses were invariably highest against NC95 at both time points. Only the NC95-G08-G08 group had higher titers against G08 at week 13. Overall, the NC95-G08-PA10 showed the broadest cross-reactivity while the NC95-PA10-PA10 and NC95-PA10-G08 groups had the highest antibody responses against most viruses as compared with the rest of the groups.

Geomean NI antibody titers against antigenically distinct influenza strains are shown in Table 4. NI antibodies were undetectable in the naïve challenge control group at both time points. In the infection-immune challenge control group (first column), titers were only high against NC95 (GMT = 139) while titers against all other viruses were ≤26. One vaccination resulted in NI antibody titers with a GMT ≥ 106 against 8 out of the 10 tested viruses in all the four infection-immune vaccinated groups (Table 4, Week 9). GMT against two human viruses (PER09 and VIC11) were ≤80. The second vaccination resulted in an overall boost of antibody responses. Remarkable were the robust antibody responses in the groups NC95-PA10-PA10 and NC95-PA10-G08 (Table 4, Week 13). Unlike for HI antibodies, there was cross-reactivity with the recent human viruses in these two groups. Cross-reactive NI antibodies against the virus belonging to the H1N2 subtype (G12) were also observed in all the vaccinated groups at both time points (GMT ≥ 106). Here again, we observed that the groups NC95-PA10-PA10 and NC95-PA10-G08 had the highest anti-NI antibody titers against most of the examined viruses.

#### 3.1.3. Protection against Challenge

To demonstrate the extent of protection against challenge, we examined virus titers in the respiratory tract as well as macroscopic and microscopic lesions of the trachea and lungs. MO15 replicated to high titers in all five pigs of the naïve challenge control group (orange bars), as shown in Figure 2. Mean virus titers were ≥10^6.1^ TCID_50_/g or 100 mg (for nasal swabs) in all tissues. In the infection-immune challenge control group (purple bars), virus was undetectable in the nasal swabs of one pig and in the lungs of two out of five pigs. Mean virus titers in this group were lower and ranged between 10^3.4^ and 10^5.9^ TCID_50_/g or 100 mg. Virus titers in the trachea and lungs were significantly lower than in the naïve challenge control group. In the four vaccinated infection-immune groups (brown, dark blue, yellow, and grey bars), the number of animals with detectable virus titers reduced and the mean virus titers were significantly reduced compared with the naïve challenge control group (*p* < 0.05). There were also significant differences in virus titers between the infection-immune challenge control group and the infection-immune vaccinated groups (*p* < 0.05), except for virus titers in nasal mucosae of the NC95-G08-G08 group and in the lungs of the NC95-G08-PA10 group (*p* > 0.05). There were no significant differences in virus titers between groups of the infection-immune vaccinated pigs, except for virus titers in nasal mucosae of the NC95-G08-G08 and the NC95-PA10-G08 groups (*p* < 0.05). Protection was more pronounced in the trachea and lungs as compared with the upper airways in all the four groups. Three pigs that received homologous vaccines (from the NC95-PA10-PA10 group) were completely protected. In the heterologous prime-boost groups, 7 of the 10 pigs were completely protected against challenge with MO15: 2 from the NC95-G08-PA10 group and all 5 pigs in the NC95-PA10-G08 group. Virus was not detected in any tissue of the NC95-PA10-G08 group.

We observed an inverse correlation between HI antibody titers at the time of challenge and virus titers in the lungs (Table 5) at 3 days post challenge.

Histopathological analysis of the lungs showed typical pathological features of influenza infection in three out of five pigs in the naïve challenge control group (Appendix A). These included bronchiolar epithelial damage, neutrophil infiltration, and alveolar wall thickening. Similar features were observed in two out of five pigs of the infection-immune challenge control group. The other three pigs had mild pathological features characterized by light peribronchiolar thickening. In three out of the four infection-immune vaccinated groups, there was no neutrophil infiltration nor lesions in the intrapulmonary airway epithelium of all pigs. However, most pigs had mild peribronchiolar thickening. The group NC95-G08-PA10, in contrast, exhibited pathological features that were similar to those of the infection-immune challenge control group. Only one pig in this group had enhanced lesions as compared to the naïve challenge control group. Taken together, the virological and histopathological analyses suggest that a prior infection induces a background immunity that can provide better protection against later infections. This immunity can be dramatically enhanced by subsequent vaccinations.

### 3.2. Antibody-Secreting and Interferon-Secreting Cell Responses after Vaccination of Infection-Immune Pigs

We also examined whether ASC and ISC responses induced by a prior infection would be enhanced following intramuscular vaccination with WIV. For this, we selected the NC95-PA10-G08 group, which showed the highest antibody responses and resulted in complete protection against challenge with an antigenically distinct strain.

#### 3.2.1. Antibody-Secreting Cell Responses

Virus-specific IgA and IgG ASC responses were examined in mononuclear cells obtained from whole blood (PBMC), TBLN, LNH, and enzymatically digested NMC. All assays were performed against the strains used for immunization (NC95, G08, and PA10) as well as two antigenically distinct human H3N2 influenza viruses, VIC75 and HK14. Figure 3 shows the numbers of IgG (Figure 3A,C,E,G) and IgA (Figure 3B,D,F,H) ASC against the five viruses in PBMC (Figure 3A,B), TBLN (Figure 3C,D), LNH (Figure 3E,F), and NMC (Figure 3G,H). All tissues from the control pigs tested negative for IgA and IgG ASC at all time points.

Infection with NC95 did not immediately stimulate ASC responses in peripheral blood and in the LNH as ASC were undetectable up until week 7 post infection (Figure 3A,B,E,F). Infection, however, stimulated the production of IgG and IgA ASC in TBLN (Figure 3C,D) and IgA ASC in NMC (Figure 3H), which were detectable at week 2. Earlier detection of ASC in these tissues is likely due to the fact that naïve B cells are dispersed throughout the respiratory airways and are thus able to respond quite rapidly to an infection [46]. At this time point, cross-reactive IgG and IgA ASC were observed in the TBLN, although responses were higher against NC95 (red dots) and the two other human influenza viruses, VIC75 (brown triangles) and HK14 (black diamonds), with mean number of cells being >25/10^6^ cells as compared with responses against the vaccine strains (blue triangles and green squares) whose means were <25/10^6^ cells. In the NMC, only IgA ASC responses were examined at week 2. Mean numbers of IgA ASC were >100/10^6^ cells against NC95, but much lower against the other four viruses (mean <50/10^6^ cells). While both IgG and IgA ASC in TBLN declined to nearly undetectable levels in most pigs by week 7, IgG and IgA ASC in the NMC increased. IgG and IgA ASC cross-reactivity was observed against all five strains in the NMC. Again, responses were higher against NC95 and the two human viruses, VIC75 and HK14, than against the vaccine strains. Vaccination at week 7 led to recall of ASC responses as seen by the increase of IgG and IgA ASC in all tissues at week 8. The responses were lowest in the LNH and highest in the NMC. The highest responses were against the virus used for infection (NC95; red dots) and the virus used for the first vaccination (PA10; blue triangles). The numbers of IgG ASC responses against NC95 in the NMC increased up to a mean of 380/10^6^ cells, while mean IgA ASC was up to 230/10^6^ cells at week 8. Cross-reactive ASC against the other three viruses were also observed, although the responses were slightly lower. Overall, IgG ASC were higher than IgA ASC responses in all tissues. At week 11, a sharp decline in numbers of both IgG and IgA ASC was observed in all tissues except the LNH, which, on the contrary, increased, suggesting a slow but gradual increase of ASC responses. The second vaccination led to re-stimulation of IgG and IgA ASC in all tissues. However, the magnitude of response after the second vaccination was lower against the viruses used for infection and first vaccination and higher against the virus used for second vaccination in all tissues.

In summary, the timing of the initial responses after infection differed between the tissues. ASC responses in TBLN and NMC occurred early after infection, while those in PBMC and LNH occurred later. Pre-existing infection-immunity against NC95 resulted in a boost of ASC responses in all examined tissues following vaccination with WIV based on antigenically distinct strains.

#### 3.2.2. Interferon-Secreting Cell Responses

Apart from examining the effects of pre-existing immunity on antibody-mediated immunity induced by vaccination, we also examined whether there were effects on cell-mediated immunity. We assessed IFN-γ-specific responses in PBMC after ex vivo stimulation with virus, as a measure of cell-mediated immunity. Figure 4 shows the numbers of ISC against each of the five viruses examined at different time points. The numbers of ISC in PBMC from the uninfected control pigs were below the detection limit of the assay at all time points. A week following infection with NC95, high numbers (mean > 50/10^6^ cells) of ISC against NC95, G08, and VIC75 were observed. Numbers of ISC against PA10 and HK14 were lower (<50/10^6^ cells). Of note were the high numbers of ISC against VIC75 (mean > 250/10^6^ cells), which is antigenically different from the virus used for infection. This cross-reactivity is in line with the notion that cross-reactive T cell responses mainly target relatively conserved internal proteins of influenza viruses instead of surface antigens [47]. The numbers of ISC declined until week 7, the time of the first vaccination. Vaccination with PA10, however, did not seem to stimulate ISC responses, except for one pig with increased numbers of ISC against four out of the five viruses, at week 8. Interestingly, the second vaccination stimulated ISC responses against all viruses except HK14, as shown by the results at week 12. The magnitude of response against all viruses except VIC75 was higher than after infection. NC95 is antigenically related to the first vaccine strain, PA10, with 91.5% amino acid homology in the HA1. On the other hand, the second vaccine strain, G08, shares only 83% amino acid homology with NC95. Studies have suggested that repeated exposure to a similar influenza strain results in limited boosting of cell-mediated immune responses [48]. This could explain why the second, but not the first, vaccination stimulated ISC responses. If this information is anything to go by, then the boosting of pre-existing CMI will be dependent on the antigenic relationship between the antigen encountered first and the antigens encountered later.

## 4. Discussion

We previously demonstrated that heterologous prime-boost vaccination with antigenically distinct H3N2 SwIAV is able to elicit broadly cross-reactive antibodies against a panel of antigenically distinct H3N2 influenza viruses of both humans and swine [17]. In that particular study, a single administration of monovalent vaccine, G08, resulted in minimal antibody responses against the vaccine strain. A booster vaccination with PA10, on the other hand, resulted in higher (≥40) antibody titers not only against the vaccine strains but against a panel of 12 out of 15 antigenically distinct H3N2 strains, and protection against challenge with either one of both strains. While this experiment was conducted in influenza-naïve pigs, most humans over the age of six have been previously infected with influenza virus [27,28]. The influence of pre-existing immunity on vaccination has been associated with both beneficial and deleterious effects [28,49,50,51,52]. In this study, we investigated the outcome of our previous heterologous prime-boost vaccination strategy with G08 and PA10 in pigs with pre-existing immunity against the human H3N2 influenza virus strain NC95.

Our results show that the background immunity induced by a previous influenza infection can increase both the breadth and the height of post-vaccination antibody responses. We could not include influenza-naïve vaccinated pigs in the present study. However, it is well known that a single administration of inactivated influenza vaccine to such pigs induces only minimal or undetectable serum HI antibody titers [17,53]. In the NC95 infection-immune pigs in contrast, the first vaccination already induced seroprotective HI titers against the vaccine strains and a number of other human and SwIAV, in addition to boosting anti-NC95 titers (Table 3, Week 9). Antibody titers were highest against the North American SwIAV that were isolated in the 1990s. We consider NC95 as the precursor of these North American SwIAV and therefore associate these responses with a back-boosting effect. In the back-boosting phenomenon, a subsequent exposure to influenza results in broad antibody responses, which are stronger against the strain encountered first than against the strains encountered later [54]. Other researchers have also reported a back-boosting effect in vaccinated individuals with pre-existing immunity. Fonville and colleagues showed that individuals with pre-existing immunity against human H3N2 viruses belonging to the Wuhan 95 cluster experienced enhanced antibody responses against Wuhan cluster viruses upon subsequent vaccination with the human A/Sydney/5/97 H3N2 virus, which belongs to a different cluster. In their case however, responses against the vaccine virus were invariably higher than those against the viruses encountered earlier [21].

Although there were differential responses between groups of infection-immune pigs, it was clear that the second vaccination resulted in an overall broadening and elevation of antibody responses as compared with responses after the first vaccination (Table 3, Week 13). Of note were the cross-reactive HI antibody titers against the two novel North American strains (MO12 and MO15) in the group NC95-G08-PA10. Vaccination of naïve pigs with G08 and PA10 in our previous study failed to induce cross-reactive antibodies against these two viruses [55]. The higher HI titers in the NC95-PA10-PA10 and NC95-PA10-G08 groups after the second vaccination as compared with the NC95-G08-G08 and NC95-G08-PA10 groups is likely due to the genetic and antigenic relationships between the strains used for immunization as well as the order of vaccine administration. It however remains unclear how these factors influence antibody responses. Throughout the experiment, the imprinting effect of NC95 was evidenced by the high anti-NC95 HI titers as compared with titers against the vaccine strains. Contrary to the concept of OAS, the induction of high anti-NC95 titers did not seem to impede the production of antibodies against other influenza strains. Other studies have also reported increased antibody responses to inactivated or computationally optimized broadly reactive antigen (COBRA) vaccines in animals primed with a live influenza virus or live attenuated influenza vaccine (LAIV) [56,57,58,59]. Francis and colleagues demonstrated that ferrets previously infected with a historical human seasonal H1N1 virus were better immunologically equipped to respond to split quadrivalent vaccine, as they were able to mount greater and longer sustained antibody responses compared with naïve-vaccinated ferrets [56]. They observed that two doses of the split quadrivalent vaccine resulted in high titers of functional antibodies that were capable of inhibiting viral infection. Using two doses of COBRA H3 HA vaccine, Allen and colleagues also demonstrated that ferrets, pre-immune to a historical human H3N2 virus, elicited HI antibodies that were broadly reactive against 12 of 13 viral variants and higher than in vaccinated naïve ferrets [59]. While these studies examined the effect of two homologous vaccines in pre-immune subjects, our study is the first to examine heterologous prime-boost vaccination.

Despite the lack of serum antibody titers against MO15 in the infection-immune challenge control group, it was partially protected against challenge with this virus strain (Figure 2). This is likely due to the induction of secretory IgA antibodies at the respiratory mucosae, which readily neutralize invading viruses and prevent them from entering into host cells [60]. Due to the polymeric structure of IgA antibodies, they are believed to be more cross-reactive as compared with the monomeric IgG antibodies [61]. These results are consistent with those of other influenza infection studies in pigs [62,63,64]. One limitation of our study is that we cannot compare protection against challenge in infection-immune vaccinated pigs with that of naïve vaccinated pigs in our previous study, since different challenge viruses were used in these two studies. However, the lack of anti-MO15 HI antibody titers in the naïve vaccinated pigs in the previous study suggests that these pigs would not be protected against challenge with MO15. The complete protection observed in one of the groups vaccinated with heterologous vaccines (NC95-PA10-G08) points to the advantages of heterologous prime-boost over homologous prime-boost vaccination strategies, given that the right combinations of vaccines strains are used.

We also aimed to examine whether ASC responses in infection-immune pigs would be enhanced following intramuscular vaccination with inactivated vaccines. We observed that infection with NC95 stimulated ASC responses in TBLN and NMC, but not in PBMC and LNH (Figure 3). Priming of naïve B cells is believed to occur in lymphoid tissues of the respiratory tract, which include mediastinal lymph nodes and mucosa-associated lymphoid tissue [65,66,67]. This could explain the lack of ASC responses in peripheral blood following infection. These naïve B cells differentiate into ASC which include plasmablasts, plasma cells, and memory B cells, with plasmablasts being the only accessible and measurable subset of cells during an acute immune response [68]. Vaccination with WIV induced ASC in PBMC and subsequently boosted responses in the TBLN and NMC. A further boost of these responses was observed in all tissues after the second intramuscular vaccination. This observation is in line with that of Jegaskanda and colleagues, who found that specific memory B cells and plasmablasts did not increase in peripheral blood after priming with LAIV [69]. However, they did increase after vaccination with inactivated vaccine. They observed that vaccination with LAIV generated robust antigen-specific germinal center B cell responses at numerous distinct sites that could be recalled following a boost with inactivated vaccine [69]. This could explain the responses that we observed in TBLN. An important limitation of this study is the lack of a naïve vaccinated group, which makes it difficult to conclude that the ASC responses observed were entirely due to pre-existing immunity. Although this group is important to make comparisons and draw conclusions, it was eliminated in the bid to reduce the number of animals used in the experiment. However, based on the increase of ASC against NC95 in the TBLN (Figure 3C,D) and NMC (Figure 3G,H) after the first vaccination (Week 8), we can confirm that there was a recall of ASC. Our study is the first to suggest that intramuscular vaccination may recall immune memory that has been induced by a prior infection with live influenza virus in the nasal mucosa. The underlying mechanisms of this booster of the mucosal immune response remain to be discovered.

Pre-existing immunity in humans is very complex because they encounter numerous influenza virus strains and vaccination throughout their lifetime. The resulting immune response is determined by the nature and genetics of the virus, the dose of virus or antigen, immunization route, and host factors such as age and immune status [18]. As such, immune repertoires and antibody landscapes will differ in different individuals, even if they are of the same age. For a better understanding of the effects of serial influenza exposures on vaccine efficacy, we would need expanded experimental studies that are designed to replicate several of the most plausible immune histories. Although natural history studies in humans are essential, they can never pinpoint exposure histories. Influenza vaccination trials in pigs may overcome some of the limitations of studies in humans and other experimental animals. Pig studies may offer a unique opportunity to answer focused questions regarding the effect of immune imprinting on vaccine efficacy. Using the pig as an influenza model, we were able to examine responses to vaccination at multiple sites of induction, which would be almost impossible in humans. Due to the large size of the pig’s tissues, we were able to examine ASC in NMC and in the lymphoid organs. Our study further supports the notion that immunologic imprinting may enhance and broaden subsequent responses to vaccinations with antigenically distinct viruses of the same HA subtype. It also suggests that infection with live virus or vaccination with LAIV may prime individuals for more robust mucosal immune responses following intramuscular vaccination with inactivated vaccines. The question as to whether infection with any other human or swine H3N2 influenza virus will lead to enhanced responses as observed in this study remains to be elucidated.

## Figures and Tables

**Figure 1 viruses-12-00968-f001:**
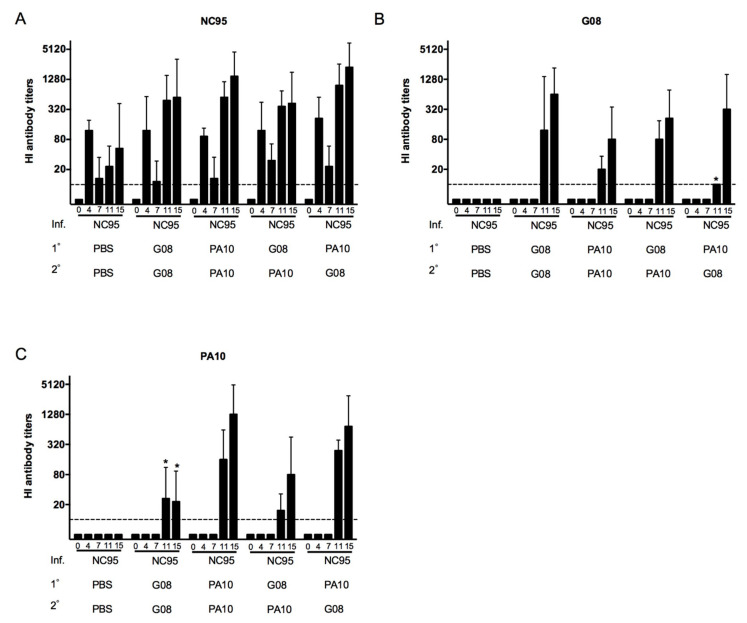
Evolution of hemagglutination inhibition (HI) antibody titers against the strains used for infection, (**A**) NC95, and vaccination, (**B**) G08 and (**C**) PA10. The titers were determined at weeks 0, 4, 7, 11, and 15 following infection with NC95. The interval between infection (Inf.) and the first vaccination (1°) was 7 weeks. The second vaccination (2°) was administered 4 weeks later. The different groups are shown on the x-axis. The dotted lines indicate the detection limit, which in this case was a titer of 10. Asterisks denote statistical differences between groups that received heterologous (NC95-PA10-G08) and homologous (NC95-G08-G08) vaccines in Mann–Whitney U test (*p* < 0.05).

**Figure 2 viruses-12-00968-f002:**
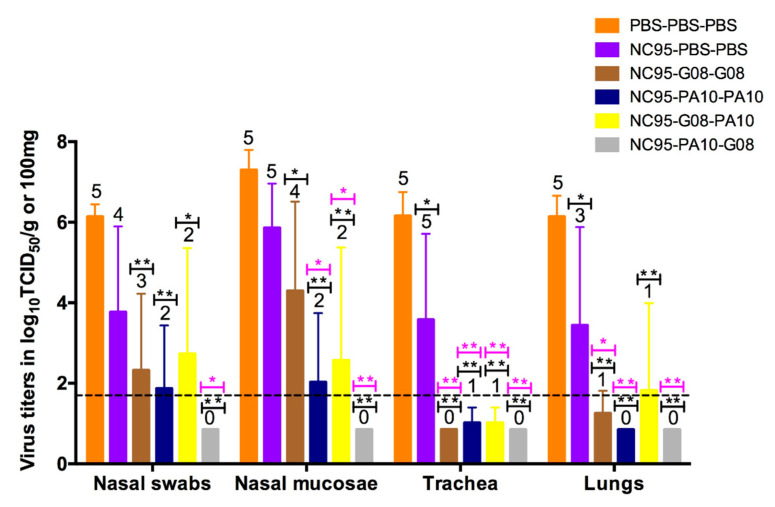
Virus titers in nasal swabs, nasal mucosa, trachea, and lungs, 3 days after challenge with 10^7^ TCID_50_ of MO15. The bars represent mean virus titers ± SD. The dotted line indicates the detection limit. Each color of the bar represents a different vaccine group as shown by the legend at the top right corner. The numbers above the bars indicate the number of pigs with detectable virus titers. Significant reductions of virus titers as compared with the naïve challenge control group (PBS-PBS-PBS, black asterisks) and infection-immune challenge control group (NC95-PBS-PBS, pink asterisks) in ANOVA are shown above the bars. * *p* < 0.05, ** *p* < 0.01.

**Figure 3 viruses-12-00968-f003:**
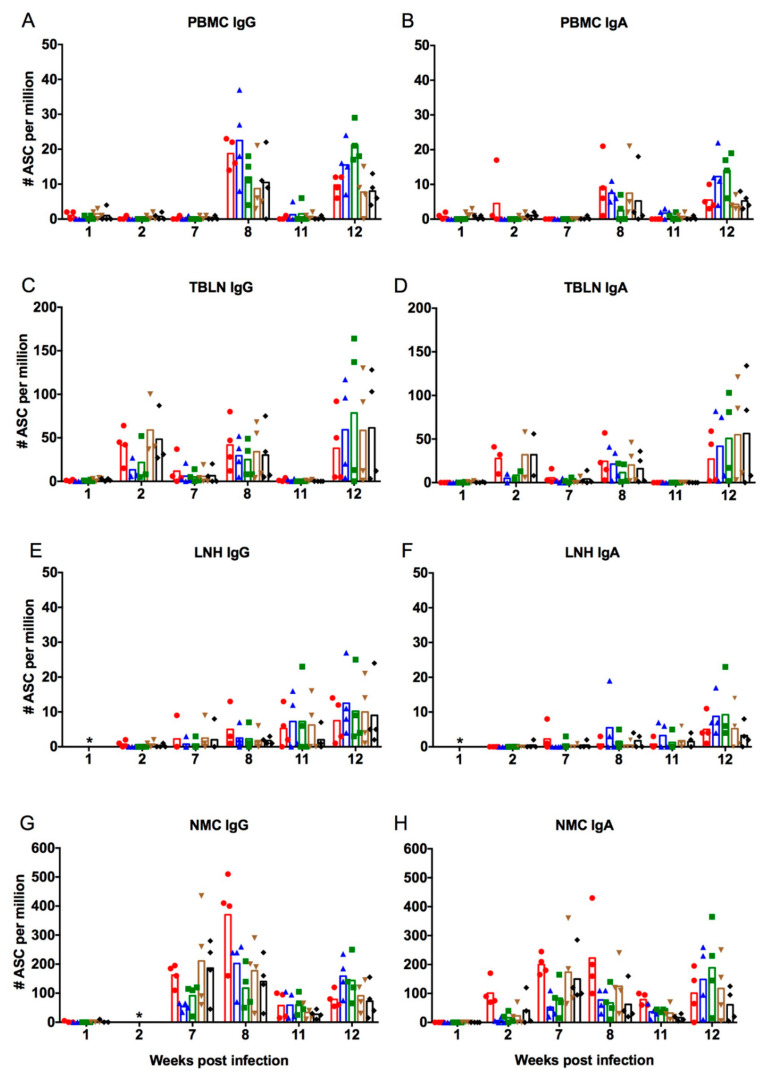
Numbers of virus-specific IgG and IgA antibody-secreting cell (ASC) in peripheral blood mononuclear cells (PBMC) (**A**,**B**), tracheobronchial lymph nodes (TBLN) (**C**,**D**), lymph nodes of the head (LNH) (**E**,**F**), and nasal mucosa (NMC) (**G**,**H**) for each individual pig after infection with NC95 and vaccination with PA10 followed by G08. Responses in the different tissues were examined at weeks 1, 2, 7, 8, 11, and 12 after infection. The first and second vaccinations were performed at weeks 7 and 11, respectively. Symbols represent responses of four individual pigs, which were examined at each time point. The mean numbers of ASC are shown by the bars. The different colors represent responses against the five different viruses (● NC95; ▲ PA10; ⯀ G08; ▼ VIC75; ◆ HK14). The asterisk (*) means that ASC responses were not examined at that particular time point.

**Figure 4 viruses-12-00968-f004:**
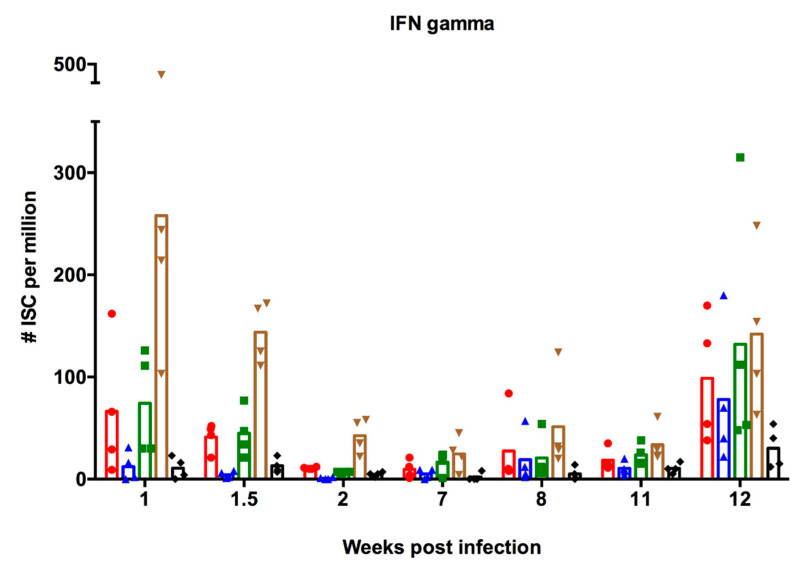
Numbers of virus-specific interferon-secreting cell (ISC) in PBMC for each individual pig after infection with NC95 and vaccination with PA10 followed by G08. Responses in PBMC were examined at weeks 1, 1.5 (day 11), 2, 7, 8, 11, and 12 after infection. The first and second vaccinations were performed at weeks 7 and 11, respectively. Symbols represent responses of four individual pigs which were examined at each time point. The mean numbers of ISC are shown by the bars. The different colors represent responses against the five different viruses (● NC95; ▲ PA10; ⯀ G08; ▼ VIC75; ◆ HK14).

**Table 1 viruses-12-00968-t001:** Genetic relationships between the HA1 of the strains used for immunization.

	Percentage Amino Acid (aa) Identities (Lower Left) and Number of Different aa at Antigenic Sites, out of a Total of 40 (Upper Right)			
	NC95	G08	PA10	MO15
NC95	-	18	12	14
G08	83.0	-	20	19
PA10	91.5	81.5	-	16
MO15	87.2	79.6	84.2	-

NC95; A/Nanchang/95, G08; A/swine/Gent/172/2008, PA10; A/swine/Pennsylvania/A01076777/2010, MO15; A/swine/A01840724/2015, -; not applicable, HA1; Hemagglutinin 1.

**Table 2 viruses-12-00968-t002:** Design of experiment 1.

Group	H3N2 Influenza Strains Used for:			
	Infection (Week 0)	Primary Vaccination (Week 7)	Booster Vaccination (Week 11)	Challenge (Week 17)
Naïve challenge control	PBS	PBS	PBS	MO15
Infection-immune challenge control	NC95	PBS	PBS	
Infection-immune vaccinated (Homologous prime-boost)	NC95	G08	G08	
	NC95	PA10	PA10	
Infection-immune vaccinated (Heterologous prime-boost)	NC95	G08	PA10	
	NC95	PA10	G08	

PBS; phosphate buffered saline, NC95; A/Nanchang/95, G08; A/swine/Gent/172/2008, PA10; A/swine/Pennsylvania/A01076777/2010, MO15; A/swine/A01840724/2015.

**Table 3 viruses-12-00968-t003:** Mean HI antibody titers against H3N2 influenza strains used for immunization and antigenically distinct strains in pigs of different vaccine groups.

		Geometric Mean HI Antibody Titers Per Group at: (Number of Pigs with Titer ≥40)									
		Week 9					Week 13				
Virus Strains	Inf.: 1°: 2°:	NC95 PBS PBS	NC95 G08 G08	NC95 PA10 PA10	NC95 G08 PA10	NC95 PA10 G08	NC95 PBS PBS	NC95 G08 G08	NC95 PA10 PA10	NC95 G08 PA10	NC95 PA10 G08
*EU swine*											
G84		5 (0)	160 (5)	53 (3)	184 (5)	46 (4)	5 (0)	1470 (5)	160 (4)	279 (5)	368 (5)
ENG87		5 (0)	15 (0)	10 (0)	17 (1)	9 (0)	5 (0)	160 (5)	30 (3)	46 (5)	23 (2)
G08		5 (0)	243 (5)	139 (5)	279 (5)	80 (5)	5 (0)	2941 (5)	485 (5)	970 (5)	1114 (5)
*N.A swine*											
TX98		13 (0)	844 (5)	640 (5)	422 (5)	1689 (5)	17 (1)	368 (5)	1470 (5)	243 (5)	1940 (5)
MN99		10 (0)	485 (5)	485 (5)	368 (5)	844 (5)	10 (0)	211 (5)	735 (5)	121 (5)	1280 (5)
ONT05		6 (0)	106 (5)	243 (5)	80 (5)	485 (5)	5 (0)	40 (3)	845 (5)	46 (3)	485 (5)
PA10		5 (0)	121 (5)	485 (5)	70 (5)	1114 (5)	5 (0)	80 (4)	4457 (5)	320 (5)	1470 (5)
IN11		5 (0)	80 (4)	184 (5)	61 (4)	279 (5)	5 (0)	121 (5)	3880 (5)	243 (4)	1940 (5)
IA11		5 (0)	368 (5)	640 (5)	368 (5)	1470 (5)	6 (0)	243 (5)	3378 (5)	422 (5)	1689 (5)
MO12		5 (0)	121 (5)	160 (5)	160 (5)	368 (5)	5 (0)	121 (5)	211 (5)	160 (5)	279 (5)
MO15		5 (0)	160 (5)	320 (5)	243 (5)	485 (5)	5 (0)	106 (5)	640 (5)	139 (5)	422 (5)
*Human*											
VIC75		5 (0)	23 (2)	13 (1)	20 (1)	20 (1)	5 (0)	121 (5)	35 (4)	48 (3)	70 (3)
ENG88		5 (0)	243 (5)	139 (5)	160 (5)	320 (5)	5 (0)	106 (5)	184 (5)	160 (5)	160 (5)
NC95		80 (5)	1689 (5)	2560 (5)	2229 (5)	4457 (5)	70 (5)	1114 (5)	5120 (5)	2560 (5)	5881 (5)
WI05		6 (0)	9 (0)	6 (0)	6 (0)	7 (0)	5 (0)	5 (0)	10 (1)	5 (0)	9 (0)
PER09		6 (0)	8 (0)	11 (1)	8 (0)	10 (1)	5 (0)	7 (0)	23 (2)	7 (0)	10 (1)
VIC11		5 (0)	5 (0)	5 (0)	5 (0)	5 (0)	5 (0)	6 (0)	7 (0)	6 (0)	12 (1)
HK14		5 (0)	6 (0)	7 (0)	5 (0)	6 (0)	5 (0)	5 (0)	35 (3)	5 (0)	7 (0)
No. of strains (n/18) against which a titer ≥40 was achieved per group		1	12	12	12	12	1	14	12	14	13

Inf., 1°, 2°: virus strains used for infection, first, and second vaccination. *N.A*: North American. HI antibody titers were determined 2 weeks after the first (week 9) and second (week 13) vaccinations. The starting dilution in the HI test was 1:10. HI antibody titers below 10 were assigned a titer of 5. Five pigs were examined per group. HI titers ≥40 are considered sero-protective. Grey indicates geometric mean HI antibody titers (GMT) titers ≥40. The color coding of the test viruses is based on antigenic clusters of the HA. Viruses of different clusters differ by at least four antigenic units. The seven human seasonal H3N2 viruses belong to six different clusters [42,43,44]. The *EU* and *N.A swine* H3N2 viruses are given the same color as their human precursors, or a different shade if they represent antigenic drift variants of swine H3N2 viruses [6,45]. Full virus names: G84; A/swine/Gent/1/1984, ENG87; A/swine/England/163266/1987, G08; A/swine/Gent/172/2008, TX98; A/swine/Texas/4199-2/1998, MN99; A/swine/Minnesota/593/1999, ONT05; A/swine/Ontario/33853/2005, PA10; A/swine/Pennsylvania/A01076777/2010, IN11; A/Indiana/08/2011, IA11; A/swine/Iowa/A01049750/2011, MO12; A/swine/Missouri/A01476459/2012, MO15; A/swine/Missouri/A01840724/2015, VIC75; A/Victoria/3/1975, ENG88; A/England/427/1988, NC95; A/Nanchang/933/1995, WI05; A/Wisconsin/67/2005, PER09; A/Perth/16/2009, VIC11; A/Victoria/361/2011, HK14; A/Hong Kong/4801/2014.

**Table 4 viruses-12-00968-t004:** Mean NI antibody titers against H3N2 influenza strains used for immunization and antigenically distinct N2 strains in pigs of different vaccine groups.

		Geomean NI Titers Per Group at:									
		Week 9					Week 13				
Virus Strains	Inf.: 1°: 2°:	NC95 PBS PBS	NC95 G08 G08	NC95 PA10 PA10	NC95 G08 PA10	NC95 PA10 G08	NC95 PBS PBS	NC95 G08 G08	NC95 PA10 PA10	NC95 G08 PA10	NC95 PA10 G08
*EU swine*											
G08		7	557	320	844	485	15	5881	1689	1940	2941
G12 *		5	557	121	106	184	10	139	735	184	368
*N.A swine*											
TX98		23	485	844	422	1470	23	279	1470	368	1114
PA10		7	640	2560	485	4457	11	485	17,829	1470	10,240
IN11		8	640	1689	640	2229	35	1689	35,658	3880	23,525
IA11		26	114	2229	1280	4457	23	557	5881	970	4457
MO15		8	243	485	243	735	9	211	1114	320	320
*Human*											
NC95		139	2941	3378	2941	6756	106	1689	5120	2560	6756
PER09		5	11	10	6	30	5	13	80	15	40
VIC11		8	40	46	23	80	7	20	70	35	121

Inf., 1°, 2°: virus strains used for infection, first, and second vaccination. NI: Neuraminidase Inhibition. NI antibody titers were determined 2 weeks after the first (week 9) and second (week 13) vaccinations. The starting dilution in the ELLA assays was 1:10. NI antibody titers below 10 were assigned a titer of 5. Five pigs were examined per group. The virus belonging to the H1N2 subtype is shown with an asterisk (*). Full virus names: G08; A/Swine/Gent/172/2008, G12; A/swine/Gent/26/2012, TX98; A/swine/Texas/4199-2/1998, PA10; A/swine/Pennsylvania/A01076777/2010, IN11; A/Indiana/08/2011, IA11; A/swine/Iowa/A01049750/2011, MO15; A/swine/Missouri/A01840724/2015, NC95; A/Nanchang/933/1995, PER09; A/Perth/16/2009, VIC11; A/Victoria/361/2011.

**Table 5 viruses-12-00968-t005:** Correlation between HI antibody titers against the challenge virus (MO15) and virus titers in the lungs.

Vaccine Group	Geomean HI Antibody Titers	Mean Lung Virus Titers (log_10_ TCID_50_/g)
PBS-PBS-PBS	5.0	6.14
NC95-PBS-PBS	5.0	3.44
NC95-G08-G08	26.4	1.25
NC95-PA10-PA10	242.5	0.85
NC95-G08-PA10	40.0	1.82
NC95-PA10-G08	160.0	0.85

HI antibody titers were determined at the time of challenge, while virus titers were determined 3 days post challenge. Correlation between titers was determined using the Spearman correlation coefficient (r = −0.91).

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
