# Peer review of "Efficacy of Heterologous Prime-Boost Vaccination with H3N2 Influenza Viruses in Pre-Immune Individuals: Studies in the Pig Model"

_viruses, 2020, doi:10.3390/v12090968_

Round 1

Reviewer 1 Report

This article by Chepkwony et al. explores the ability of prime-boost immunization with inactivated intramuscular vaccines to induce broadly reactive antibodies against group 2 influenza A viruses in pigs with pre-existing immunity. This article is executed and written well. My suggestions to improve the article are as follows: 

  1. Did the researchers observe OAS or antigenic seniority in pigs? Explaining antibody response from this perspective would be helpful for future investigations using pig as a model of OAS or immune history studies. Comparing antibody responses after vaccination (e.g. on week 15) against NC95, G08 and PA10 in different experimental groups, in figure 1, may answer this.
  2. Explain the color coding used for antigenic relatedness of virus strains used in table 3. It does not make senses to ask readers to go through another article to figure out antigenic relatedness of viruses used in this research.
  3. It is necessary to have couple of sentences in result section explaining why the particular experimental group (NC95-PA10-G08) was selected for further analysis of ASCs and ISCs. The way antibody data are presented does not explain whether there were any differences among the vaccine groups.
  4. It would be better to include the correlation plot as well in the article. Also, was there any correlation with NI titer and protection? Can the authors determine the protective level of antibodies in this study?
  5. Was there any specific reason why the challenge virus-specific ASCs and ISCs were not determined?
  6. Lack of no infection and prime-boost vaccination control in ASCs and ISCs study makes it harder to make sure that the increment in these cells are due to pre-existing infection-induced immunity. The authors need to make it clearer in the article.
  7. Further explanation of how this can be translated in human vaccination and what would be the future experimental approach in pig model is needed. 

Reviewer 2 Report

Viruses-907446: Efficacy of heterologous prime-boost vaccination with H3N2 influenza viruses in per-immune individuals: studies in the pig model.

In this original research, Chepkwony et al. infect pigs with 1 virus, vaccinate with different influenza virus (or viruses) to look at antibody responses, as well as infect with yet another virus to look at protection from a distinct virus. While this topic is of importance, as it is common for multiple influenza viruses to be circulating at once within the same pig farm, the authors need to revise the manuscript so that the readers can follow the data. For example, the authors will use different terminology in the text vs. what is on the figure to explain results. Additionally many figures contain so much data and the authors do not point the reader to what to look at in the text (figures A, B, etc. compare colors, etc.).

General comment: There are no line numbers, which makes it difficult to review for specific comments. The template comes with line numbers, which should be left in. Please make sure these line numbers are provided in the resubmission.

Specific Comments:

Methods: The sentence about most people over the age of 30 have been exposed to NC95 needs a reference.

Figures 1 and : Each graph should be labeled (A, B, etc.) and when discussing in the text, each figure should be referenced so the reader knows which panel to look at.

3.1.1. The authors spend a lot of space describing the fold changes in antibody responses, but for many of these no statistics are done/significantly different.

3.1.2. For both paragraphs (each describing a different table), the paragraphs end abruptly. What is the overall summary of the data/what should the reader take home?

            For both tables, the list of full virus names should also be put with their appropriate abbreviations.

            The paragraph describing Table 4 needs to be expanded for clarification. For example, talking about cross-reactivity with recent human viruses-what is the reader supposed to be looking at in the table (need virus names, specific columns/weeks, etc.). Additionally, the last sentence refers to cross-reactivity of H1N2 subtype, which looking at the table it looks like there isn’t one and the author needs to read in the table legend-the authors should change G12 being under  the current EU Swine H3N2 column and distinguish it is H1N2.

3.1.3. Third line needs Figure 2 reference at the end of the sentence.

Figure 2 and corresponding text is difficult to follow. There is a lot of data, which makes it tough, but the text uses the phrases “infection-immune challenge control” vs. “infection immune vaccination” whereas the table does not use these terms (instead viruses), making it difficult to follow. Since most of the statistical significance is from the trachea and lung samples, it would be easier for the reader (and this reviewer) to follow if the bars were grouped together by sample location, with the bar colors being the different infections. Then the authors can use their different wording in the text vs. the figure, but then guide where the reader should look. For ex. 2 different pig sets “…in the lungs” (Black bar vs white bar). Then the reader knows to look at the lung cluster of bars and can compare and not have to go back to look at what the phrases mean as far as viruses/PBS. It will also make showing the statistics easier as well.

Histopathological analysis: Table S1 is reference, but no supplemental tables/information is provided and these data cannot be reviewed.

Section 3.2.1. This is a large figure with a lot of data and the reviewer/reader cannot follow. The figure needs panels and the text needs to reference where the reviewer/reader need to be directed to. Once this is redone the reviewer will look at and critique the data/text, etc.

Figure 4 is hard to read with the data being squished (symbols of one virus overlap with another virus). The authors should break the Y axis and make the graph wider so it is legible.

Discussion: Please provide figure references from this study when making conclusions.

The sentence that starts with “Francis and colleges demonstrated…” needs a reference.

Round 2

Reviewer 1 Report

Thanks for the revised article addressing all the comments. Would have been easier if the authors rechecked the line numbers, there were mismatches in author response and manuscript.